# Vasorelaxant Activity of *Salvia hispanica* L.: Involvement of the Nitric Oxide Pathway in Its Pharmacological Mechanism

**DOI:** 10.3390/molecules28176225

**Published:** 2023-08-24

**Authors:** Luis A. Herbert-Doctor, Amanda Sánchez-Recillas, Rolffy Ortiz-Andrade, Emanuel Hernández-Núñez, Jesús Alfredo Araujo-León, Tania Isolina Coral-Martínez, Nubia Noemi Cob-Calan, Maira Rubi Segura Campos, Samuel Estrada-Soto

**Affiliations:** 1Laboratorio de Farmacología, Facultad de Química, Universidad Autónoma de Yucatán, Mérida 97069, Yucatan, Mexico; luis.herbert@correo.uady.mx (L.A.H.-D.); amanda.sanchezrecillas@gmail.com (A.S.-R.); 2Departamento de Recursos del Mar, Centro de Investigación y de Estudios Avanzados del IPN, Mérida 97310, Yucatán, Mexico; emanuel.hernandez@cinvestav.mx; 3Unidad de Bioquímica y Biología Molecular de Plantas, Centro de Investigación Cinetífica de Yucatán, A.C., Mérida 97205, Yucatan, Mexico; jalfredoaraujo@gmail.com; 4Laboratorio de Cromatografía, Facultad de Química, Universidad Autónoma de Yucatán, Mérida 97069, Yucatan, Mexico; tcoral@correo.uady.mx; 5Instituto Tecnológico Superior de Calkiní en el Estado de Campeche, Calkiní 24900, Campeche, Mexico; nubia.cob@itconkal.edu.mx; 6Facultad de Ingeniería Química, Universidad Autónoma de Yucatán, Mérida 97203, Yucatán, Mexico; maira.segura@correo.uady.mx; 7Facultad de Farmacia, Universidad Autónoma del Estado de Morelos, Cuernavaca 62209, Morelos, Mexico; enoch@uaem.mx

**Keywords:** *Salvia hispanica*, ^13^C NMR dereplication, nitric oxide pathway, vasorelaxant

## Abstract

*Salvia hispanica* L., commonly known as *chía*, and its seeds have been used since ancient times to prepare different beverages. Due to its nutritional content, it is considered a dietary ingredient and has been reported with many health benefits. Chia seed components are helpful in cardiovascular disease (CVD) by reducing blood pressure, platelet aggregation, cholesterol, and oxidation. Still, its vasodilator effects on the vascular system were not reported yet. The hexanic (HE*Sh*), dichloromethanic (DE*Sh*), and methanolic (ME*Sh*) extracts obtained from *chía* seeds were evaluated on an aortic ring *ex-vivo* experimental model. The vasorelaxant efficacy and mechanism of action were determined. Also, phytochemical data was obtained through ^13^C NMR-based dereplication. The ME*Sh* extract showed the highest efficacy (E_max_ = 87%), and its effect was partially endothelium-dependent. The mechanism of action was determined experimentally, and the vasorelaxant curves were modified in the presence of *L*-NAME, ODQ, and potassium channel blockers. ME*Sh* caused a relaxing effect on KCl 80 mM-induced contraction and was less potent than nifedipine. The CaCl_2_-induced contraction was significantly decreased compared with the control curve. Phytochemical analysis of ME*Sh* suggests the presence of mannitol, previously reported as a vasodilator on aortic rings. Our findings suggest NO-cGMP pathway participation as a vasodilator mechanism of action of *S. hispanica* seeds; this effect can be attributed, in part, to the mannitol presence. *S. hispanica* could be used in future research focused on antihypertensive therapies.

## 1. Introduction

*Salvia* L. genus is the great abundant taxonomic group of the Lamiaceae family with ca. 1000 species [1]. The *Salvia* spp., have a wide worldwide distribution [2] and has been used in ancient traditional medicine, as food, and even in the production of cosmetics in different parts of the world [3,4,5], e.g., *Salvia hispanica* L., commonly known as *chía*, is an essential Mesoamerican plant. Since ancient times, *chía* seeds have been used to prepare different beverages such as “*agua fresca de chía*”, a drink mixed with lemon and sugar, or “*atole*”, a drink mixed with toasted maize [6,7], and foods such as “*chiapinolli*” flour, roasted and grinding seeds [8].

*Chía* seeds are considered in the European Union as a food ingredient for their nutritional content [9]. It has been employed in the food industry in bakery, dairy, meat, and fish products [10]. The nutritional components of *chía* comprise high content of soluble and insoluble fiber, protein, minerals, vitamins, and saturated fatty acids, principally from the ω-3 group, such as α-linolenic acid [11,12]. Also, different secondary metabolites such as caffeic acid, chlorogenic acid, quercetin [8], myricetin, oleacein [13] and rosmarinic acid [14] have been reported as principal components. Currently, research on *chía* seeds on the antioxidant [15,16], anti-diabetic [14,17] weight loss [18] and anti-hypertensive [19,20] biological activities are of great interest considering the clinical benefits on cardiovascular risk. These characteristics target to *chía* seeds as a potential functional food, principally by health-promoting based on the *in vitro* and *in vivo* reports to have cardiovascular protection effects [15]. Some functional foods, such as *chía* have antioxidant properties and thus are helpful in prevents of cardiovascular disease (CVD) by reducing blood pressure, platelet aggregation, cholesterol, and oxidation [21]. Chia seeds contain omega-3 fatty acids and antioxidants that can be used as a functional component to aid in reducing the risk of CVD, thus, considering that the Mexican population has a prevalence of cardiovascular disease and that *chía* is one of our country’s pseudo cereals of high consumption, the current study aimed to explore the potential of Mexican *chía* seeds as a functional food helpful vascular disease.

Nitric oxide (NO) is a small gas molecule that has been found in the vascular endothelium layer. It plays a critical physiological role in maintaining the homeostasis of blood vessels, including the regulation of vascular tone [22]. The mechanisms underlying the regulation of vascular tone by NO are well understood. In the vascular endothelium, the amino acid L-arginine is used constitutively by the endothelial type of NO synthase (eNOS) to produce NO, which diffuses into adjacent smooth muscle cells to promote cGMP formation and subsequently, cGMP leads to vasodilation [23]. 

In the current work, we present for the first time the vasorelaxant effect of *S. hispanica* seeds in an *ex vivo* experimental model to identify its cardiovascular benefits. In addition, an approximation of the mechanism of action and a phytochemical analysis are presented.

## 2. Results

### 2.1. Extraction Yield

Following the maceration extraction of *S. hispanica* seeds, the percentage of major yield was obtained from the hexanic extract (HE*Sh*, 16.75%) followed by dichloromethanic (DE*Sh*, 2.5%) and methanolic (ME*Sh*, 1.5%) extracts, respectively. The three extracts were evaluated using an aortic ring pre–contracted with norepinephrine (NE) to determine their efficacy as important vasorelaxant agents in cardiovascular diseases.

### 2.2. Pharmacological Evaluation 

#### 2.2.1. Vasorelaxant Effect

The ME*Sh* extract was the most effective and potent test sample in endothelium–intact aortic rings (E+) (E_max_ = 87.69%; CE_50_ = 124 μg/mL) compared with the HE*Sh* (E_max_ = 28%; CE_50_ = >1000 μg/mL) and DE*Sh* (E_max_ = 14%; CE_50_ = >1000 μg/mL) (Figure 1a). Whereas without endothelium the vasorelaxant effect was not significant (Figure 1b). The ME*Sh* vasorelaxant effect was concentration and partially endothelium–dependent, while HE*Sh* and DE*Sh* did not showed significant efficacy. In both cases, any extract was more active than positive controls (carbachol E+: E_max_ = 85% or nifedipine E–: 100%) (Figure 1a,b). Table 1 shows pharmacological data of E_max_ and EC_50_ values, calculated using extracts and positive controls.

#### 2.2.2. Mechanism of Action

The ME*Sh* showed the major vasorelaxant effect, so its mechanism of action was assessed. The ME*Sh* effect was partially endothelium dependent. This suggests that endothelium–derived relaxant factors (EDRF) such as nitric oxide (NO), prostacyclin (PGI_2_), or Endothelium–Derived Hyperpolarizing Factor (EDHF) and direct mechanisms on vascular smooth muscle cells as antagonism of the adrenergic receptor or calcium channel blocking are involved in the ME*Sh*–induced relaxation [24]. Based on earlier data, the participation of EDRF as NO and PGI_2_ was first investigated.

The ME*Sh* extract showed the major vasorelaxant effect, so its mechanism of action was assessed. A ME*Sh* relaxation curve was shifted to the right in the presence of *L*–NAME (E_max_ = 18.48%; EC_50_ = >1000 μg/mL) and ODQ (E_max_ = 28.32%; EC_50_ = >1000 μg/mL) and efficacy was significantly lowered (Figure 2a), while relaxation curve in the presence of indomethacin or atropine were not altered respect to the control (ME*Sh*: E_max_ = 87.69%; CE_50_ = 124 μg/mL) (Figure 2b). The curve in the presence of potassium channel blockers such as 2–aminopyridine, glibenclamide, and tetraethylammonium were significantly modified (Figure 3a). The same shows the relaxant curve of ME*Sh* in aortic rings contracted with high potassium solution (KCl; 80 mM) (Figure 3b). The ME*Sh* extract was efficient (E_max_ = 70%) compared with a calcium channel blocker (nifedipine, E_max_ = 99%). While Figure 3c shows the CaCl_2_–induced contraction curve. The ME*Sh* significantly caused inhibition of calcium contraction as well as nifedipine. The efficacy of norepinephrine in the presence of ME*Sh* was not modified. 

#### 2.2.3. Phytochemical Analysis

The ^13^C NMR–based dereplication was performed by comparing the chemical shift (*δ*_C_) experimental values from the crude extracts (Appendix A) with those predicted chemical shifts contained in a dedicated database Salvia/DB. Based on the score obtained from comparing the experimental vs. predicted chemical shift, the dereplication analysis performed with HE*Sh* and DE*Sh* extract suggested the presence of alcohols, saturated and unsaturated fatty acids in the first 20 results (Table 2 and Table 3), of which linoleic acid (18/18 matching carbons, score 1) was confirmed in HE*Sh* by comparing their experimental chemical shift vs. reported chemical shift [25] (Appendix A). Similarly, the results of the ME*Sh* extract show the presence of sugars, glycosides, and polyalcohols (Table 4); of these, mannitol (6/6 matching carbons, score 1) was confirmed [26] (Appendix A). The presence of fatty acids in DE*Sh* was not confirmed.

## 3. Discussion

Many research groups are investigated the nutraceutical properties and pharmacology efficacy of *S. hispanica* as an antioxidant [16], anti–diabetic [14,17], a weight loss adjuvant [18], and hypertensive prevent [19,20]. Thus, this work was focused to obtained evidence of the vasorelaxant direct effect and phytochemistry composition of *S. hispanica* seed. Three extracts of *S. hispanica* seeds were prepared: HE*Sh*, DE*Sh*, and ME*Sh*, and their preclinical pharmacology efficacy were tested. The ME*Sh* extract showed more efficiency, and vasorelaxant efficacy was partially endothelium–dependent, so its mechanism of action was attributed to a dual mechanism correspondent to endothelium factors’ presence and targets in smooth muscle cells [24,27].

The role of the vascular endothelium is affected by the presence of membrane-bound receptors like cholinergic, mainly M_3_ receptors, which when activated by an agonist induces nitric oxide production and release [28]. The ME*Sh* curve in the presence of atropine (an M_3_ antagonist) was not significantly modified. Thus, behavior discarded M_3_ receptor activation as the first trigger of the extract vasorelaxant effect. The major endothelium–derived releasing factor (EDRF) is nitric oxide (NO), which diffuses to smooth muscle cells where the guanylyl cyclase enzyme is activated, thus catalyzing GMP to cGMP conversion. To identify the participation of nitric oxide pathway activation, endothelium–intact aortic rings were pre-incubated with *L*–NAME (an NO synthetase inhibitor) or ODQ (a guanylyl cyclase inhibitor), respectively, and after that, the ME*Sh* relaxant curve was obtained. Both curves were lowered in the presence of these inhibitors, thus suggesting that the NO–cGMP pathway could be involved in ME*Sh*–induced relaxation [29]. On the other hand, it is well known that PGI_2_ endothelium releases can evoke a vasorelaxant effect [30]. Thus, with the purpose to identified prostaglandins–participation in the ME*Sh* effect, aortic rings were pre–incubated with indomethacin (an unspecific prostaglandins inhibitor); however, the ME*Sh* efficacy and potency were not modified, thus discarding PGI_2_–participation in ME*Sh* vasorelaxant effect [30].

As described, EDRF can increase the second’s messengers as cGMP or cAMP and thus cause PKG or PKA activation, which participates in the opening of KATP, potassium channel calcium–activation (K_Ca_), and repolarization. In consequence, the relaxation process is evoked [31]. In this context, the relaxant effect of ME*Sh*, in the presence of tetraethylammonium (K_Ca_ channels blocker), glibenclamide (an ATP–sensitive potassium channel blocker), and or 2–aminopyridine (an inhibitor of KV), was significantly decreased by a half percent (Figure 3a), and the curve was shifted to the right respect to control (indicating a loss in potency), suggesting potassium channels opening because of NO-production ME*Sh* induced. The opening of potassium channels will cause cell repolarization, and consequently, calcium channel blocking occurs [32].

To assess if Ca^2+^ channel blockade was involved in the vasorelaxant effect of the ME*Sh*, the assay was carried out in Ca^2+^–free of RKH solution, and a curve for contraction with CaCl_2_ was obtained (control curve). The contractile effect induced by CaCl_2_ was compared in the absence and presence of the ME*Sh* (EC_50_ = 124.7 μg/mL). The CaCl_2_-induced contraction was significantly decreased by MES*h*-like nifedipine used as the positive control (Figure 3c). Moreover, the ME*Sh* (3.03 to 1000 μg/mL) produced a moderate vasorelaxant effect on the contraction induced with high potassium Krebs solution (E_max_ = 65%) (Figure 3b). These behaviors suggest a half blocker of L–type Ca^2+^ channels in the membrane or due to the repolarization process [33]. Changes in the intracellular Ca^2+^ concentration and membrane depolarization stimulate large–conductance Ca^2+^–activated K^+^ (BKCa^2+^) channels, which are thought to play an essential role in maintaining the membrane potential of vascular smooth muscle cells [34].

Meanwhile, phytochemical analyses were performed through the ^13^C–NMR dereplication method as a spectroscopic strategy that has proven to help identify NPs in complex mixtures, even without purification [35]. This technique offers advantages over the dereplication analyses using mass spectrometry (MS), liquid chromatography (LC), gas chromatography (GC), or combined instrumentation such as LC–UV, LC–MS, or GC–MS [36]. Here, we have used ^13^C–NMR chemical shift values obtained from crude extracts for the dereplication process to obtain a phytochemical approximation of *S. hipanica* seeds obtained in Jalisco, México. Based on the comparison of experimental vs. reported chemical shifts (Appendix A), the results showed the presence of linoleic acid (Table 2), previously reported in aerial parts of *Salvia triloba* L. f., [37]; and mannitol (Table 4), reported in leaves of *S. officinalis* [38]. This is the first report of mannitol on *S. hispanica* seeds. Also, the endothelium–dependent vasorelaxant effect of mannitol on mousse mesenteric arterioles was reported and attributed to hyperosmotic action. Therefore, this compound could be responsible for ME*Sh* vasorelaxant effect [38]. However, further studies are needed to determine the presence of mannitol and its relationship with the reported biological activity.

## 4. Materials and Methods

### 4.1. Plant Material and Extraction

Seeds of *Salvia hispanica* L. (ID: SF2G/90/2022) were purchased with certified supplies (SuperFoods2Go) in Jalisco, México, in June 2022. Seeds were dried and ground at room temperature (25 °C) for 3 days. Powdered seeds (400 g) were submitted to successive maceration processes (three times, 24 h each) using 100% of: *n*–hexane, dichloromethane, and methanol (100 g/L), correspondingly. After filtration through a Whatman grade 1 filter, organic extracts were concentrated under reduced pressure using a rotary evaporator (BUCHI R-300) coupled to a vacuum pump (Lab Companion VE-11) and lyophilized using CentriVap FriZowe 6 (LABCONCO) to give the correspondent low (HE*Sh*, 67 g), medium (DE*Sh*, 10 g) and high (ME*Sh*, 6 g) polarity extracts.

### 4.2. Chemicals and Solution Preparation

(+/-)-norepinephrine bitartrate hydrate (NE), carbamylcholine chloride (carbachol), indomethacin, *L*-NG-Nitroarginine methyl ester (*L*-NAME), 1H-[1,2,4]Oxadiazolo[4,3-a]quinoxalin-1-one (ODQ), potassium chloride (KCl), calcium chloride (CaCl_2_), and dimethylsulfoxide (DMSO) were purchased from Sigma-Aldrich Co. (St. Louis, MO, USA).

Krebs Henseleit Solution; composition, mM: NaCl, 119; KCl, 4.6; KH_2_PO_4_, 1.2; MgSO_4_, 1.2; CaCl_2_, 1.5; NaHCO_3_, 20; EDTA, 0.026 and glucose, 11.4.

All other reagents and solvents were analytical grade. Stock solutions of all the chemicals were made in distilled water, except for the extracts, which were dissolved in DMSO (10%). Fresh dilutions were made at the day of the experiment.

### 4.3. Experimental Subjects

Adult male Wistar rats (250–300 g bodyweight) were obtained from the Universidad Anáhuac–Mayab animal house in Mérida, Yucatan, México. Animals were housed in polycarbonate cages and maintained under standard laboratory conditions (12–h light/dark cycle, 25 ± 2 °C and humidity 45–65%) and were fed with a standard rodent diet and water *ad libitum*. All animal procedures were conducted by Secretaría de Agricultura, Ganadería, Desarrollo Rural, Pesca y Alimentación of México (SAGARPA, 1999) [39] and approved by the Institutional Animal Care and Use Committee (UJAT-0001-2017). All experiments were carried out using six animals per group. All study animals were sacrificed by cervical dislocation after deep anesthesia with phenobarbital (65 mg/kg, i.p.).

### 4.4. Pharmacology Assay

#### 4.4.1. General Experimental Procedures

The Adult male Wistar rats were sacrificed accordingly to the method described by Sánchez-Recillas et al. (2020) [40]. Cervical dislocation and thoracic dissection were carried out to extract the thoracic aorta. It was cleaned from adjacent and connective tissue and then cut into strips 3 mm long (we use a Vernier measurement instrument). In addition, for some aortic rings, the endothelium layer was gently removed by cotton rub manual procedures. The aortic rings were assembled in chambers at 37 °C containing Krebs-Henseleit solution (KHS) at pH 7.4 using stainless steel hooks under an optimal tension of 3 g. After that, aortic rings were submitted to stabilize period for 20 and were constantly bubbled with an O_2_:CO_2_ (95:5%) mixture. Changes in tension were recorded by force transducers Grass-FT03 (Astromed, West Warwick, RI, USA) connected to an analyzer MP-150 (BIOPAC 4.1 Instruments, Santa Barbara, CA, USA).

A sensitization process was carried out after the stabilization period. The tissues were stimulated with noradrenaline (NE, 0.1 μM) for 15 min after washed with fresh KHS, and allowed to stabilize for 15 min, this procedure was repeat three times. The absence of endothelium layer was confirmed by the lack of the relaxant response (>50%) induced by carbachol (CCH; 1 μM) in the last contraction with NE before washing with fresh KHS to assess viability.

#### 4.4.2. *Ex Vivo* Vasorelaxant Evaluation

After sensitization period, the tissues were allowed to stabilize for 20 min and then, contracted with NE (0.1 μM). Extracts (3.03 to 1000 μg/mL), vehicle (100% final concentration) or the positive controls CCH for endothelium–intact aortic rings (E+; 0.303 to 100 μg/mL) and nifedipine for endothelium–denuded aortic rings (E-; 3.89 × 10^−5^ to 3.46 μg/mL) were added to the chamber in quarter–logarithm dilutions and cumulative concentration–response curves (CRC). The relaxant effect of the samples was determined by their ability to reduce the maximal vascular contraction (E_max_) and potency (EC_50_) effects induced by NE comparing the tissue tension before and after their addition.

#### 4.4.3. Mechanism of Action Approach

In order to establish the mechanism of action of methanol extract of *S. hispanica* (ME*Sh*), the following experiments were conducted.

(a) To establish a possible antagonism of adrenergic receptors or disruption of the NE pathway, the following procedures were performed on E– aortic rings. A cumulative NE-induced contraction (4.15 × 10^−11^ to 3.6 × 10^−5^ M) of CRC was made as the positive control (control CRC). In another experiment, aortic rings were pre–incubated with ME*Sh* (EC_50_ = 125 μg/mL) for 15 min. Then the CRC to NE-induced contraction was performed to compare the contraction induced by NE in the absence and presence of ME*Sh*.

(b) To know the role of endothelium-derived relaxing factors as nitric oxide (NO) or prostacyclin (PGI_2_), the E+ aortic rings were pre-incubated with NG-nitro-L-arginine methyl ester (*L*–NAME, nitric oxide synthase inhibitor (100 μM) or indomethacin and cyclooxygenases inhibitor (10 μM), respectively for 15 min, before the contraction with NE (0.1 μM). The relaxation CRC of ME*Sh* (3.03 to 1000 μg/mL) was built as described in the vasorelaxant set experiments. The maximal relaxing effect of the ME*Sh* was compared in the absence and presence of *L*–NAME or indomethacin, respectively.

(c) To establish the possible inhibition of soluble guanylyl cyclase enzyme (sGC), the E+ aortic rings were pre–incubated with 1-H-[1,2,4]-oxadiazolo-[4,3a]-quinoxalin-1-one (ODQ an sGC inhibitor (10 μM) for 15 min, previous to the contraction with NE (0.1 μM). The relaxation CRC of ME*Sh* (3.03 to 1000 µg/mL) was built as described before. The maximal relaxing effect of the ME*Sh* was compared in the absence and presence of ODQ.

(d) To know the role of K^+^ channels on extract–induced vasorelaxant effect, the E+ aortic rings were pre–incubated with tetraethylammonium (TEA, non–selective K_Ca_ channels blocker (10 μM), 2–Aminopyridine (2AP; 100 μM) an inhibitor of voltage-gated potassium channels (KV) or glibenclamide (10 μM) an ATP–sensitive potassium channel blocker (K_ATP_) for 15 min, previous to the contraction with NE (1 µM). The relaxation CRC of ME*Sh* (3.03 to 1000 µg/mL) was built as described before. The maximal relaxing effect of the ME*Sh* was compared in the absence and presence of K_ATP_.

(e) To determine whether inhibition of extracellular Ca^2+^ influx was involved in the extract-induced vasorelaxation, the experiments were carried out in Ca^2+^–free KHS. After sensitization, endothelium–intact aortic rings were washed with Ca^2+^–free KHS containing KCl (80 mM) and stabilized for 15 min. Then, a CRC for CaCl_2_–induced contraction was obtained without the ME*Sh* (control group). Once the maximal contraction was reached, tissue was washed with Ca^2+^–free, KCl (80 mM), and KHS, and allowed to stabilize for 20 min. Finally, after 15 min incubation with the ∫ (EC_50_ = 125 μg/mL), another CRC for CaCl_2_-induced contraction was obtained. The contractile effect induced by CaCl_2_ was compared in the absence and presence of the ME*Sh*.

### 4.5. Phytochemical Analysis

#### 4.5.1. Natural Products Databases for ^13^C NMR Dereplication

A database (DB) of natural products (NPs) for ^13^C NMR–based dereplication process was prepared with some modifications to the method described by Bruguière et al. (2021) [41]. Briefly, a search was carried out for previously NPs reported in *Salvia* spp., using LOTUS: Natural Products Online, available at https://lotus.naturalproducts.net/ (accessed on 20 March 2023). The structures resulting from the LOTUS research were exported in .sdf format, containing 2392 NPs. The ^13^C NMR chemical shifts (*δ*_C_) were predicted for each NPs’ methyl, methylene, methine, and quaternary carbons using the algorithm described by Nuzillard (2021) [42] and the ACD NMR predictor (ACD/Labs) to obtain Salvia/DB in the required format for MixONat software (v.1.0.1., SONAS, France).

#### 4.5.2. ^13^C NMR–Based Dereplication

Dereplication analyses were carried out using MixONat software available at https://sourceforge.net/projects/mixonat/ (accessed on 24 April 2023) [43]. Previously, 50 mg of the crude plant extracts were dissolved in chloroform–*d*4 or methanol–*d*4 (600 µL). Carbon spectra (^13^C–NMR, 150 MHz) were obtained on a Varian-Agilent AR Premium Compact spectrometer (Santa Clara, CA, USA). The spectra were acquired with 6000 scans, and the spectral width was 230 ppm. Phase and baseline correction of spectra was performed automatically using MestReNova software (v. 12.0.0, Mestrelab Research, Santiago de Compostela, Spain). A minimum intensity threshold was then used to collect positive ^13^C–NMR signals. Each spectrum’s experimental *δ*_C_ list and intensities data were exported to a .csv file using Excel Microsoft Office (Microsoft, Redmond, WA, USA) as an input file in MixONat. The software ranked the putative NPs contained in the mixture with a range score between 1 and 0, according to the number of matching experimental *δ*_C_ obtained from extracts vs. the predicted *δ*_C-SDF_ values of NPs in the Salvia/DB. The predicted NPs were then confirmed by comparing the experimental vs. with that *δ*_C_ reported in the literature analyzed in the same deuterated solvent, with a displacement tolerance range of ±0.5 ppm.

### 4.6. Statistical Analysis

The results are expressed as the standard error of the mean (*n* = 5) ± SEM. Concentration-response curves (CRC) were plotted, and the experimental data from the CRC were adjusted by the nonlinear Hill equation with a curve-fitting program (ORIGIN 8.0 MICROCAL). The statistical significance of differences between means was assessed by a one-way analysis of variance (ANOVA) followed by Tukey’s post hoc test; *p*-values lower less than 0.05 (* *p* < 0.05) were statistically significant [44,45].

## 5. Conclusions

The vasorelaxant effect of *S. hispanica* seems to be characterized, and our results suggest NO–cGMP pathway participation as a vasodilator mechanism of action. Also, the preliminary phytochemical report was presented. This work is the first report on *ex vivo* pharmacology analysis of *S. hipanica* species, which could be used in future research focused on antihypertensive therapies.

## Figures and Tables

**Figure 1 molecules-28-06225-f001:**
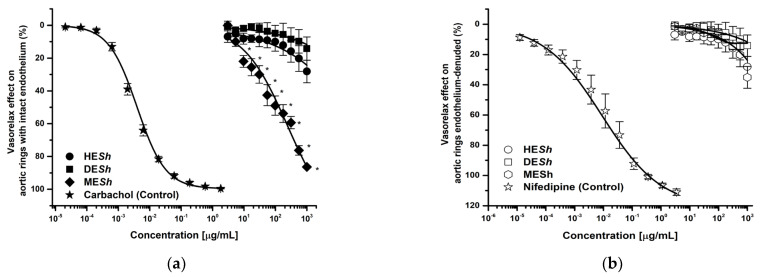
Concentration–response curves of vasorelaxant effect of *S. hispanica* extracts on intact–endothelium (**a**) and denuded–endothelium aortic rings (**b**). Results are expressed as the means ± SEM of six experiments, * *p* < 0.05 represents a statistically significant difference between extracts. HE*Sh*: hexane extract DE*Sh*: Dichloromethane extract, ME*Sh*: methanol extract. Carbachol: positive control in endothelium–intact aortic rings, Nifedipine: positive control in endothelium–denuded aortic rings.

**Figure 2 molecules-28-06225-f002:**
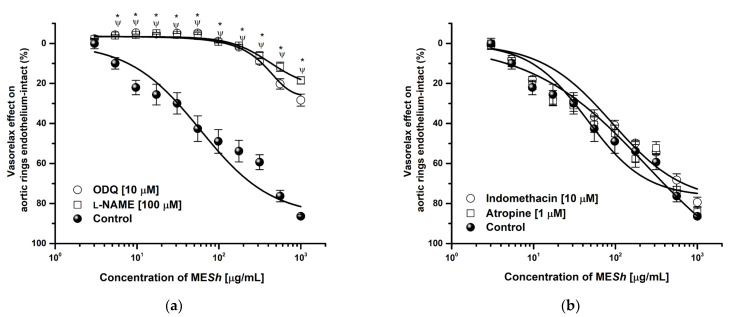
Concentration–response curves of vasorelaxant effect of ME*Sh* in the presence of (**a**) *L*-NAME (100 μM) an ODQ (10 μM), (**b**) Indomethacin (10 μM) and Atropine (1 μM) on intact-endothelium. Results are expressed as the means ± SEM of six experiments, * *p* < 0.05 represents a significant difference compared with the control.

**Figure 3 molecules-28-06225-f003:**
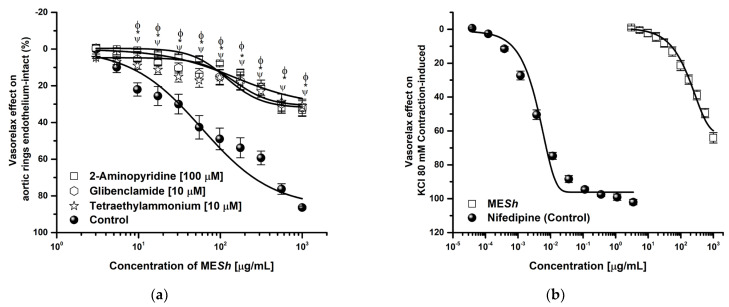
Concentration–response curves of (**a**) vasorelaxant effect of ME*Sh* in presence of potassium channel blockers: 2–Aminopyridine (100 μM), glibenclamide and tetraethylammonium (10 μM), (**b**) vasorelaxant effect of ME*Sh* On high potassium KCl (80 mM) –induced contraction and (**c**) effect of ME*Sh* on CaCl_2_–induced contraction. Results are expressed as the means ± SEM of six experiments, * *p* < 0.05 represents significant difference compared with control group.

**Table 1 molecules-28-06225-t001:** Pharmacological data of the vasorelaxant effect of *S. hispanica* extracts and control drugs.

Sample Test	E_max_ (%)	EC_50_ (μg/mL)
HE*Sh* (E+)	28.12 ± 6.91	>1000 ^+^
HE*Sh* (E–)	11.45 ± 2.2	>1000 ^+^
DE*Sh* (E+)	14.19 ± 7.1	>1000 ^+^
DE*Sh* (E–)	12.34 ± 3.5	>1000 ^+^
ME*Sh* (E+)	**87.69 ± 0.78**	**124.74 ***
ME*Sh* (E–)	35.1 ± 7.19	>500 ^+^
Carbachol (control E+)	99.03 ± 0.23	0.0041
Nifedipine (control E–)	100 ± 2.15	0.0033

HE*Sh*: hexanic extract, DE*Sh*: dichloromethane extract, ME*Sh*: methanol extract, E+: endothelium-intact aortic rings, E–: endothelium-denuded aortic rings, Emax: maximum effect, EC_50_: median effective concentration. * Indicates significant statistical difference compared with HE*Sh* and DE*Sh*, *p* < 0.05 ANOVA one way and *pos hoc* Tukey. *^+^* Parameter not determined since the maximum effect was less than 50%.

**Table 2 molecules-28-06225-t002:** First 20 NPs predicted from the results of the ^13^C NMR–based dereplication analysis, of the HE*Sh* extract.

Rank	Name	Score (*δ*_C_ Match)
1	Nonanoic acid	1.0 (9/9 carbons)
2	Hexanol	1.0 (6/6 carbons)
3	Capryc acid	1.0 (10/10 carbons)
4	Undecanoic acid	1.0 (11/11 carbons)
5	Heptanoic acid	1.0 (7/7 carbons)
6	Amyl alcohol	1.0 (5/5 carbons)
7	Hexanoic acid	1.0 (6/6 carbons)
8	Undecane	1.0 (11/11 carbons)
9	Octanol	1.0 (8/8 carbons)
10	2-hexenol	1.0 (6/6 carbons)
11	3-octenol	1.0 (8/8 carbons)
12	Caprylic acid	1.0 (8/8 carbons)
13	Myristoleic acid	1.0 (14/14 carbons)
14	Linoleic acid	0.1 (18/18 carbons)
15	A-linolenic acid	0.1 (18/18 carbons)
16	Arachidonic acid	0.95 (19/20 carbons)
17	Methyl linolenate	0.95 (18/19 carbons)
18	Methyl octadeca-9,12,15-trienoate	0.95 (18/19 carbons)
19	Hexyl octanoate	0.93 (13/14 carbons)
20	Lauric acid	0.92 (11/12 carbons)

**Table 3 molecules-28-06225-t003:** First 20 NPs predicted from the results of the ^13^C NMR–based dereplication analysis, of the DE*Sh* extract.

Rank	Name	Score (*δ*_C_ Match)
1	Undecane	1.0 (11/11 carbons)
2	Nonanoic acid	1.0 (9/9 carbons)
3	Hexanol	1.0 (6/6 carbons)
4	N-tridecanoic acid	1.0 (13/13 carbons)
5	Amyl alcohol	1.0 (5/5 carbons)
6	Capric acid	1.0 (10/10 carbons)
7	Lauric acid	1.0 (12/12 carbons)
8	Caprylic acid	1.0 (8/8 carbons)
9	Undecanoic acid	1.0 (11/11 carbons)
10	Myristoleic acid	1.0 (14/14 carbons)
11	Hexyl octanoate	1.0 (14/14 carbons)
12	Octanol	1.0 (8/8 carbons)
13	Myristic acid	1.0 (14/14 carbons)
14	Oleic acid	0.1 (18/18 carbons)
15	Arachidonic acid	0.1 (20/20 carbons)
16	Methyl linolenate	0.95 (18/19 carbons)
17	Linoleic acid	0.94 (17/18 carbons)
18	A-linolenic acid	0.94 (17/18 carbons)
19	Pentadecanoic acid	0.93 (14/15 carbons)
20	Methyl octadeca-9,12,15-trienoate	0.89 (17/19 carbons)

**Table 4 molecules-28-06225-t004:** First 20 NPs predicted from the results of the ^13^C NMR–based dereplication process, of the ME*Sh* extract.

Rank	Name	Score (*δ*_C_ Match)
1	Alpha-L-glucopyranose	1.0 (6/6 carbons)
2	Beta-L-glucopyranose	1.0 (6/6 carbons)
3	Mannitol	1.0 (6/6 carbons)
4	Sucrose	1.0 (12/12 carbons)
5	Caprylic acid	1.0 (8/8 carbons)
6	Hexyl acetate	0.88 (7/8 carbons)
7	(2r,3s,4s,5r,6s)-2-(hydroxymethyl)-6-{[(1s,4r,6s)-1,3,3-trimethyl-2-oxabicyclo[2.2.2]octan-6-yl]oxy}oxane-3,4,5-triol	0.88 (14/16 carbons)
8	(2r,3s,4s,5r,6s)-2-(hydroxymethyl)-6-{[(1s,4r,6r)-1,3,3-trimethyl-2-oxabicyclo[2.2.2]octan-6-yl]oxy}oxane-3,4,5-triol	0.88 (14/16 carbons)
9	2-(hydroxymethyl)-6-({1,3,3-trimethyl-2-oxabicyclo[2.2.2]octan-6-yl}oxy)oxane-3,4,5-triol	0.88 (14/16 carbons)
10	5-methylhexanoic acid	0.86 (6/7 carbons)
11	Heptanoic acid	0.86 (6/7 carbons)
12	Hexyl octanoate	0.86 (12/14 carbons)
13	Βeta-D-glucopyranose	0.83 (5/6 carbons)
14	Βeta-D-glucosa	0.83 (5/6 carbons)
15	2-hexenol	0.83 (5/6 carbons)
16	Hexanol	0.83 (5/6 carbons)
17	Hexanoic acid	0.83 (5/6 carbons)
18	Agropine	0.82 (9/11 carbons)
19	Undecanoic acid	0.82 (9/11 carbons)
20	(2r,3s,4s,5r,6s)-2-(hydroxymethyl)-6-{[(1r,4s,6r)-1,3,3-trimethyl-2-oxabicyclo[2.2.2]octan-6-yl]oxy}oxane-3,4,5-triol	0.81 (13/16 carbons)

## Data Availability

The data presented in this study are available in Appendix A.

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
