# Peer review of "Vasorelaxant Activity of Salvia hispanica L.: Involvement of the Nitric Oxide Pathway in Its Pharmacological Mechanism"

_molecules, 2023, doi:10.3390/molecules28176225_

Round 1
Reviewer 1 Report
Dear authors, I was given the opportunity to review your article: Vasorelaxant Activity of Salvia hispanica L.: Involvement of the Nitric Oxide Pathway in its Pharmacological Mechanism. I really like the idea and idea with Salvia (chia). It is a possibility to significantly contribute to the molecular efficiency of the cardiovascular system - you have dealt with the role of NO. At the same time, I highlight the use of the right inhibitors. At the same time, I highlight a lot of work done (I know exactly what it entails to measure vascular reactivity for 18 hours a day non-stop for 3 months). That's why I have many comments, questions and requests for editing, as well as supplementing either already measured data or completing experiments so that what you are going to present to the scientific community with your work is clear and justifiable.
One of my initial requests will be to complete the picture with the exact experimental procedure. Consolidate a bit of the entire work into a more comprehensive whole. The Introduction needs to be supplemented and reworked. There is a lack of basic information about NO and vasoconstriction-relaxation properties, why it is measured and what we expect from it. At the same time, in Discussion, it does not respond to the Introduction, on the other hand, it is a description of the Result. A deep change is needed. Consistency is lacking. From point A to point B. The results are partially satisfactory. The material and methods need to be supplemented and expanded. For example, if I were to repeat your research, I am unable to do so. I don't even know how to prepare HESh, DESh, and MESh solutions. I don't know what devices and procedures you used. If it is important to refer to your former publications.
I see one fact as a significant shortcoming. You are talking about the bias tension of the vascular segment at 0.5g and 3g. Neither of these factors is common when using vascular reactivity measurement. Justify. In case you can't justify, I ask you to supplement the experiments with a vascular pre-tension of 1g - that is, the tension that is at atmospheric pressure. You crossed it three times. I would not apply this preload even with hypertension. At higher tension, spontaneous constriction of the vascular segment occurs, disruption of the homeostasis of relaxation-constriction properties. Speaking of segments, how did you achieve the exact 3mm cut? That's just from the edge, because cutting according to a ruler seems quite ... passable.
You are talking about mannitol. So let's look at this molecule. I did not notice the concentration of mannitol in the MESh sample you determined. I went through several published papers. Of course, you have to look at the fact that YOU have to work on the article, not that I will now spend weeks looking for answers. I did not directly find a publication where the relaxing effect of mannitol on the endothelium is shown, only that it has a relaxing effect. But this is important.
Due to the fact that the pictures are not standardized in some way, you communicate the effect of carbachol only in the MESh group without endothelium. What about the control group.
Maybe everything I write is completely fine, but without clear information, I don't know and I can't confirm that. Therefore, I will ask for a clear evaluation of your research. I could be wrong too, but at least you can see how many questions I have, and that's only because I'm missing step by step.
Non-published materials and supplementary are the same components. Is this a mistake? It's not the same! Unpublished data is still unpublished.
Finally, one small thing. Please reconsider your use of the ex vivo connection as I have looked at over 30 publications and they all use in vitro that I was aware of. Then I really don't know what and how you measured if it was ex vivo
Minor revision
I would like to change name - Luis A. Herbert, not doctor - because I found the Internet without this verbal connection
Line 100: you omitted e.g. H2S, what about its verification?
Line 117: enter non-pubicated data in the supplement
Line 223: I completely understand that the temperature is different in Mexico. But room temperature 35°?
Line 243: permission number?
Line 252: what procedures was the endothelium removed?
Line 261-263: I cannot agree with the procedure. I would like to complete the experiment when the sequence will be: adjustment of tension, deendothelialization, effect of carbachol, administration of the substance, carefully check the effect of carbachol. The reason: never if I try the effect of acetylcholine at the beginning, then study the molecule and try acetylcholine again, I don't know anything about the effect of the substance. At the same time, carbachol and acetylcholine should always be tested first.
Line 338: I just learned that Bailey and Daniel and Cross introduced mean and SEM. Please edit and at the same time I have some questions about statistics. Why was Two-way ANOVA not used, as your results are necessary for this type of statistical evaluation. I like that you used non-linear regression, but it would be appropriate to add tangent lines to the curves. The directive in this case determines the molecular effect.
Please check for errors and typos, for example, I found inconsistent MESh notations - sometimes it is in italics and sometimes not
I request a grammar, spelling and error check
Author Response
Author’s response to the review letter
Dear reviewers, we sincerely appreciate each of your comments, which undoubtedly provided valuable feedback to enrich and improve our manuscript. Below, we have listed each of your comments and provided a specific response to address them. We have also made the necessary and relevant changes in the document in accordance with each comment.
Reviewer 1
Dear authors, I was given the opportunity to review your article: Vasorelaxant Activity of Salvia hispanica L.: Involvement of the Nitric Oxide Pathway in its Pharmacological Mechanism. I really like the idea and idea with Salvia (chia). It is a possibility to significantly contribute to the molecular efficiency of the cardiovascular system - you have dealt with the role of NO. At the same time, I highlight the use of the right inhibitors. At the same time, I highlight a lot of work done (I know exactly what it entails to measure vascular reactivity for 18 hours a day non-stop for 3 months). That's why I have many comments, questions and requests for editing, as well as supplementing either already measured data or completing experiments so that what you are going to present to the scientific community with your work is clear and justifiable.
One of my initial requests will be to complete the picture with the exact experimental procedure. Consolidate a bit of the entire work into a more comprehensive whole. The Introduction needs to be supplemented and reworked. There is a lack of basic information about NO and vasoconstriction-relaxation properties, why it is measured and what we expect from it. At the same time, in Discussion, it does not respond to the Introduction, on the other hand, it is a description of the Result. A deep change is needed. Consistency is lacking. From point A to point B. The results are partially satisfactory. The material and methods need to be supplemented and expanded. For example, if I were to repeat your research, I am unable to do so. I don't even know how to prepare HESh, DESh, and MESh solutions. I don't know what devices and procedures you used. If it is important to refer to your former publications.
Response: We appreciate the comments, however we kindly comment that: We were not measuring vascular reactivity for 18 h. so it was a drafting mistake, now we reviewed the first manuscript and we add methodology details.
Our work was realized with the methodology employed and standardized previously by Sánchez-Recillas A, Navarrete-Vázquez G, Hidalgo-Figueroa S, Bonilla-Hernández M, Ortiz-Andrade R, Ibarra-Barajas M, Yáñez-Pérez V, Sánchez-Salgado JC. (2020) Pharmacological characterization of the cardiovascular effect of Nibethione: ex vivo, in vivo and in silico studies. Journal of Pharmacy and Pharmacology [published online ahead of print, 2020 Jun 4]. DOI:10.1111/jphp.13295.
Also, the introduction was supplemented with NO-antecedents and its vascular importance.We add methodological details of the test of the vasorelaxant effect.
We add methodological details of extraction process.
I see one fact as a significant shortcoming. You are talking about the bias tension of the vascular segment at 0.5g and 3g. Neither of these factors is common when using vascular reactivity measurement. Justify. In case you can't justify, I ask you to supplement the experiments with a vascular pre-tension of 1g - that is, the tension that is at atmospheric pressure. You crossed it three times. I would not apply this preload even with hypertension. At higher tension, spontaneous constriction of the vascular segment occurs, disruption of the homeostasis of relaxation-constriction properties. Speaking of segments, how did you achieve the exact 3mm cut? That's just from the edge, because cutting according to a ruler seems quite ... passable.
Response: We use the methodology reported in: Sánchez-Recillas A, Navarrete-Vázquez G, Hidalgo-Figueroa S, Bonilla-Hernández M, Ortiz-Andrade R, Ibarra-Barajas M, Yáñez-Pérez V, Sánchez-Salgado JC. (2020) Pharmacological characterization of the cardiovascular effect of Nibethione: ex vivo, in vivo and in silico studies. Journal of Pharmacy and Pharmacology [published online ahead of print, 2020 Jun 4]. DOI:10.1111/jphp.13295.
We were not evaluated the “vascular reactivity”, we determinate vasorelaxant effect in this same the initial tension in aortic rings was 3g, after we evoked norepinephrine contraction then test samples vasorelaxant activity were determinates. We used this value because the methodology vasorelaxant has been standardized in our research group and previous experiments has been reported with this experimental method:
This link there are other references: https://scholar.google.com.mx/scholar?q=ex+vivo+vasorelaxant+effect&hl=es&as_sdt=0&as_vis=1&oi=scholart
You are talking about mannitol. So let's look at this molecule. I did not notice the concentration of mannitol in the MESh sample you determined. I went through several published papers. Of course, you have to look at the fact that YOU have to work on the article, not that I will now spend weeks looking for answers. I did not directly find a publication where the relaxing effect of mannitol on the endothelium is shown, only that it has a relaxing effect. But this is important.
Due to the fact that the pictures are not standardized in some way, you communicate the effect of carbachol only in the MESh group without endothelium. What about the control group.
Response: Please note that we are using here a 13C-NMR dereplication as a strategy for a first phytochemical approach of extracts. This technique analyzes the experimental chemical shifts obtained from the sample and compares them to a predicted of natural products (NPs) chemical shift contained in a database. Then the tentative predicted NPs are verified with the reported chemical shifts, however it is not possible to determine the concentration of these NPs in a complex mixture.
Maybe everything I write is completely fine, but without clear information, I don't know and I can't confirm that. Therefore, I will ask for a clear evaluation of your research. I could be wrong too, but at least you can see how many questions I have, and that's only because I'm missing step by step.
Response: We appreciate the comments, We have considered all your questions and comments and have improved the manuscript. Thank you for taking the time to review it.
Non-published materials and supplementary are the same components. Is this a mistake? It's not the same! Unpublished data is still unpublished.
Response: The material was included as Supplementary.
Finally, one small thing. Please reconsider your use of the ex vivo connection as I have looked at over 30 publications and they all use in vitro that I was aware of. Then I really don't know what and how you measured if it was ex vivo.
Response: ex vivo word is employed as a type of in vitro assay but ex vivo is specifically use of tissue isolated (on the spot) and experiments gotten in conditions like to natural physiology. Examples Below:
-Mariana Torres-Piedra, Mario Figueroa, Oswaldo Hernández-Abreu, Maximiliano Ibarra-Barajas, Gabriel Navarrete-Vázquez, Samuel Estrada-Soto,
Vasorelaxant effect of flavonoids through calmodulin inhibition: Ex vivo, in vitro, and in silico approaches,Bioorganic & Medicinal Chemistry,Volume 19, Issue 1,2011, Pages 542-546, ISSN 0968-0896,
https://doi.org/10.1016/j.bmc.2010.10.063.
- Gabriela Avila-Villarreal, Oswaldo Hernández-Abreu, Sergio Hidalgo-Figueroa, Gabriel Navarrete-Vázquez, Fabiola Escalante-Erosa, Luis M. Peña-Rodríguez, Rafael Villalobos-Molina, Samuel Estrada-Soto, Antihypertensive and vasorelaxant effects of dihydrospinochalcone-A isolated from Lonchocarpus xuul Lundell by NO production: Computational and ex vivo approaches, Phytomedicine, Volume 20, Issue 14, 2013,
Pages 1241-1246, ISSN 0944-7113,
https://doi.org/10.1016/j.phymed.2013.06.011.
- Maria Yolanda Rios, Sugey López-Martínez, Fabian López-Vallejo, José L. Medina-Franco, Rafael Villalobos-Molina, Maximiliano Ibarra-Barajas, Gabriel Navarrete-Vazquez, Sergio Hidalgo-Figueroa, Oswaldo Hernández-Abreu, Samuel Estrada-Soto, Vasorelaxant activity of some structurally related triterpenic acids from Phoradendron reichenbachianum (Viscaceae) mainly by NO production: Ex vivo and in silico studies, Fitoterapia, Volume 83, Issue 6, 2012, Pages 1023-1029, ISSN 0367-326X,
https://doi.org/10.1016/j.fitote.2012.05.014.
Y this link there are other references: https://scholar.google.com.mx/scholar?q=ex+vivo+vasorelaxant+effect&hl=es&as_sdt=0&as_vis=1&oi=scholart
Minor revision
I would like to change name - Luis A. Herbert, not doctor - because I found the Internet without this verbal connection
Response: Please note that the complete name of the first author is Luis Alfredo Herbert-Doctor. Doctor is his second last name.
Line 100: you omitted e.g. H2S, what about its verification?
Response: The vasorelaxation induced by H2S comprises a minor endothelium-dependent effect and a major direct effect on smooth muscles, which differs from the effects of NO and CO that act only on smooth muscles. While activation of the cGMP pathway is an important mechanism for NO- and CO-induced vasorelaxation, the H2S-induced vasorelaxation is mediated mainly by the opening of KATP channels in vascular SMCs. Our results showed that the main mechanism of action of the MESh is dependent on endothelium presence, so we focus the experimental methodology to identify FRDE participation on the other hand, actually we have not an H2S inhibitors in our lab. We appreciate the comment, and we consider implementing this experiment in future research.
-Zhao W, Zhang J, Lu Y, Wang R. The vasorelaxant effect of H(2)S as a novel endogenous gaseous K(ATP) channel opener. EMBO J. 2001 Nov 1;20(21):6008-16. doi: 10.1093/emboj/20.21.6008. PMID: 11689441; PMCID: PMC125693.
Line 117: enter non-publicated data in the supplement
Response: The material was included as Supplementary
Line 223: I completely understand that the temperature is different in Mexico. But room temperature 35°?
Response: We appreciate the comment. Please note that the extracts were obtained in Merida, Yucatan, Mexico, here the minimum and maximum temperature average are 24 ºC and 36 ºC, respectively. However, we have maintained the plant material under controlled temperature at 25 ºC; we modified this information in the manuscript.
Line 243: permission number?
Response: UJAT-0001-2017
Line 252: what procedures was the endothelium removed?
Response: The procedure was included in the manuscript.
Line 261-263: I cannot agree with the procedure. I would like to complete the experiment when the sequence will be: adjustment of tension, deendothelialization, effect of carbachol, administration of the substance, carefully check the effect of carbachol. The reason: never if I try the effect of acetylcholine at the beginning, then study the molecule and try acetylcholine again, I don't know anything about the effect of the substance. At the same time, carbachol and acetylcholine should always be tested first.
Response:
We employed a reported previously methodology of vasorelaxant effect and mechanism of action determination:
Below the methodology is summarized:
The vasorelaxant experiment was realized in consecutive steps:
1.- The tissue was obtained.
In this step, aorta is dissected and cleaned and it’s cut in segments of 3mm long, we measure with Vernier instrument. Clean and cut procedures are realized with aorta inside Krebs solution (every moment).
2.- Remove endothelium; After, some aortic rings are rubbing with a cotton stylet to remove endothelium layer, this process is confirmed after.
3.- Sensitization process; This experimental process is realized to know aorta integrity, endothelium presence and receptors sensitization induce. In this step we induced norepinephrine contraction and we measure maximum relaxant effect of carbachol. The aortic rings without endothelium layer were not respond to carbachol.
4.-Vasorelaxant effect; This step is realized after sensitization in only viable aortic rings. We induce contraction with norepinephrine when maximal contraction is obtained, extracts and controls are added to incubation chamber in concentration increase order manner.
Line 338: I just learned that Bailey and Daniel and Cross introduced mean and SEM. Please edit and at the same time I have some questions about statistics. Why was Two-way ANOVA not used, as your results are necessary for this type of statistical evaluation. I like that you used non-linear regression, but it would be appropriate to add tangent lines to the curves. The directive in this case determines the molecular effect.
Response:
4.6. The Statistical Analysis Section was edited.
Thank you for your comment, in fact you are right, the ideal would be to use Two-way ANOVA to compare between extracts, however only one extract (MESh) was active (Emax=87%) and only one variable was considerate (concentration in vasorelaxant effect or inhibitor presences in mechanism of action experimental process ) for this reason one-way ANOVA was used.
Please check for errors and typos, for example, I found inconsistent MESh notations - sometimes it is in italics and sometimes not.
Response: We appreciate the comments. Please note that now, we are uniformized the abbreviation in the manuscript.
Reviewer 2 Report
Abstract
Line 23 – Three extracts from chia… (Describe the three extracts in the abstract)
Introduction
Reflecting to the title. No describe on the role of nitic oxide (NO) in the cardiovascular disease and why the NO is the main subject of study…please describe in intro?
Methodology
Line 223- how many days the seeds were dried?
Line 224- How many percent of concentration for each solvent was used? 100% hexane? 100% dichloromethane and 100% methanol.
Line 226- an evaporator or Rotary evaporator was used.
Line 237: Rephrase – Universidad Anahuac-Mayab animal house.
Line 242- please provide an Animal Ethical Approval reference number.
Line 248- rats were scarified….
Line 252- describe in detail the manual procedures.
Line 226- why chose this range of concentration 3.03-1000 ug/mL? Please justify.
Line 343- typo error : were statistically..not ‘statically’
Author Response
Reviewer 2
Abstract
Line 23 – Three extracts from chia… (Describe the three extracts in the abstract).
Response: We appreciate the comments. Please note that now, we are mentioned in the abstract the obtention of the three chía extracts.
Introduction
Reflecting to the title. No describe on the role of nitic oxide (NO) in the cardiovascular disease and why the NO is the main subject of study…please describe in intro?
Response: Nitric Oxide antecedents were included in manuscript.
Methodology
Line 223- how many days the seeds were dried?
Response: We appreciate the comments. Please note that we are mentioned now the days that the seeds where dried.
Line 224- How many percent of concentration for each solvent was used? 100% hexane? 100% dichloromethane and 100% methanol.
Response: We appreciate the comments. Please note that we are now included the percent of concentration for each solvent used for obtain the plant extract.
Line 226- an evaporator or Rotary evaporator was used.
Response: We appreciate the comments. Please note that we are now mentioned that a rotary evaporator was used to concentrate the solvent of the correspondent plant extract.
Line 237: Rephrase – Universidad Anahuac-Mayab animal house.
Response: We appreciate the comments. We have changed it.
Line 242- please provide an Animal Ethical Approval reference number.
Response: We appreciate the comments. We have included it.
Line 248- rats were scarified….
Response: We appreciate the comments. We have changed it.
Line 252- describe in detail the manual procedures.
Response: We appreciate the comments. This information was included.
Line 226- why chose this range of concentration 3.03-1000 ug/mL? Please justify.
Response: The maximum concentration employed commonly of many research groups is 500 to 1000 mcg/mL in vegetal extracts pharmacology screening. You can see below:
-Aguilar-Guadarrama, A.B.; Yáñez-Ibarra, G.; Cancino-Marentes, M.E.; González-Ibarra, P.; Ortiz-Andrade, R.; Sánchez-Recillas, A.; Rodríguez-Carpena, J.-G.; Aguirre-Vidal, Y.; Medina-Diaz, I.-M.; Ávila-Villarreal, G. Chromatographic Techniques and Pharmacological Analysis as a Quality Control Strategy for Serjania triquetra a Traditional Medicinal Plant. Pharmaceuticals 2022, 15, 1289. https://doi.org/10.3390/ph15101289
-Pérez-Barrón G, Estrada-Soto S, Arias-Durán L, Cruz-Torres KC, Ornelas-Mendoza K, Bernal-Fernández G, Perea-Arango I, Villalobos-Molina R. (2022) Calcium Channel Blockade Mediates the Vasorelaxant Activity of Dichloromethane Extract from Roots of Oncidium cebolleta on Isolated Rat Aorta. Biointerfaceresearch 13 (1) https://doi.org/10.33263/BRIAC131.080
-Hernández-Abreu O, Castillo-España P, León-Rivera I, Ibarra-Barajas M, Villalobos-Molina R, González-Christen J, Vergara-Galicia J, Estrada-Soto S. (2009) Antihypertensive and vasorelaxant effects of tilianin isolated from Agastache mexicana are mediated by NO/cGMP pathway and potassium channel opening. Biochemical Pharmacology, 1, 54-61.
- Estrada-Soto S, González-Trujano M E, Rendón-Vallejo P, Arias-Durán L, Ávila-Villarreal G, Villalobos-Molina R. (2021) Antihypertensive and vasorelaxant mode of action of the ethanol-soluble extract from Tagetes lucida Cav. aerial parts and its main bioactive metabolites, Journal of Ethnopharmacology, 266, 113399. https://doi.org/10.1016/j.jep.2020.113399.
Response: Previous studies were realized of our research group and the vasorelaxant method was standardized with ten dilutions starting 1000 mcg/mL, thus 1000, 560, 313.6, 175.62, 98.34, 55.07, 30.84, 17.27, 9.67, 5.42, 3.03 (mcg/mL) this concentration range was used in other experimental data reported, besides we can see vasorelaxant effect concentration-dependent in detail.
Line 343- typo error : were statistically..not ‘statically’
Response: We appreciate the comments. Please note that we are now fixed the error from “statically significant” to “statistically significant”.
Round 2
Reviewer 1 Report
Dear authors. Thank you for your edits, the work gained a much simpler picture and understanding of the scientific intent. Nevertheless, I still have comments and let's try to find a reasonable way so that I don't harm you, or that I release work where I disagree with the opinion.
Important reminder:
I'm very sorry, but I have to say it again. I disagree with the 3g bias. (something else is 3g/3mm). So, the co-author is also Mrs. Sánchez-Recillas A. I had to go to the very beginning of her citations and publications.
SRA cites Journal of Pharmacy and Pharmacology, Volume 72, Issue 9, September 2020, Pages 1186–1198, https://doi.org/10.1111/jphp.13295, who are cite BOLETÍN LATINOAMERICANO Y DEL CARIB DE PLANTAS MEDICINALES Y AROMÁTICAS 17 (3): 310 - 323 (2018) © / ISSN 0717 7917 /; who are cite J Pharm Pharmacol. 2010 Sep;62(9):1167-74. doi: 10.1111/j.2042-7158.2010.01146.x.; who are cite Life Sci. 2006 Aug 8;79(11):1062-8. doi: 10.1016/j.lfs.2006.03.006. Epub 2006 Apr 21.; who are cite Aguirre-Crespo, F., Castillo-España, P., Villalobos-Molina, R., López-Guerrero, J.J., Estrada-Soto, S., 2005. Vasorelaxant effect of Mexican medicinal plants on isolated rat aorta. Pharmaceutical Biology 43 (6), 540–546; who are cite Villalobos-Molina R, Ibarra M (1996) a-Adrenoreceptorsmediating contraction in arteries of normotensive andspontaneously hypertensive rats are of the a1D or a1Asubtypes. Eur J Pharmacol 298: 257–263; who cite themselves and their foundations https://doi.org/10.1016/0014-2999(95)00074-U; https://doi.org/10.1016/0014-2999(93)90195-N; DOI: 10.1016/0014-2999(93)90195-n; 10.1016/s0922-4106(05)80019-7; ... and so on.
Somewhere in the quotation, there is a mention of preload 2 and 3g. However, if we focus on the animal type, we go from rabbit aorta to rat aorta. The first difference. The second difference is that I went through other publications and then used the citations of the author SRA. I found a line where he refers to previous work and at once I get to the 1g bias, as it should be.
Therefore, the question arises. Not on the basis of which publication, but on the basis of which scientific outputs was the bias set to 3g? I still think it's a lot, compared to 1-1.5g preload.
That is why I propose a consensus. Add either in the title or in the methods that it is an increased vascular pretension, but it would be appropriate to cite a publication that directly deals with tension in the vessel and differences in vasorelaxant and vasoconstrictor responses.
If this fact is pointed out, which you also emphasize in the limitation section of the manuscript, I am fully able to accept this fact.
175-176 line : I am in favor of editing the text: we have control and add MESh. Relaxation looks to be 100% .... that is, MESH by way of ENDOTHELIUM/SMOOTH MUSCLE, the vascular segment relaxed.
By adding L-NAME or ODG, relaxation was present at 20%.
Thus, the words "modification" and "may" are incorrect, but "sensitivity to the cumulative addition of MESh, after inhibition of the NA and sGC pathway, was reduced" (F2). At the same time, you observed that, unlike HESh and DESh, with MESh relaxation takes priority (as evidenced by a de-endothelialized vascular segment and a reduced ability to relax by 80%)
Line 190: How is curve 3a shifted to the right? Those are the tangents in the E50 that I mentioned last time
Line 234-235: You write about the use of the substances n-hexane, dichloromethane, and
methanol, but no control of the pure substance, without active leached substances, is available in the publication or in the supplement. Either I didn't understand it, but you let the prepared substances evaporate. However, when there was, for example, methanol, was there no concentration of methanol in the resulting substance MESh? It would be good to emphasize and justify this, because, for example, methanol can react with the vascular segment. And you used 100% solutions according to the materials.
small edits:
line 212: MESh
line 268: camera or voltage sensor?
There are still shortcomings, avoid guesswork, use clear wording of answers. Correct typos.
Author Response
REVIEWER 1
Important reminder:
I'm very sorry, but I have to say it again. I disagree with the 3g bias. (something else is 3g/3mm). So, the co-author is also Mrs. Sánchez-Recillas A. I had to go to the very beginning of her citations and publications.
SRA cites Journal of Pharmacy and Pharmacology, Volume 72, Issue 9, September 2020, Pages 1186–1198, https://doi.org/10.1111/jphp.13295, who are cite BOLETÍN LATINOAMERICANO Y DEL CARIB DE PLANTAS MEDICINALES Y AROMÁTICAS 17 (3): 310 - 323 (2018) © / ISSN 0717 7917 /; who are cite J Pharm Pharmacol. 2010 Sep;62(9):1167-74. doi: 10.1111/j.2042-7158.2010.01146.x.; who are cite Life Sci. 2006 Aug 8;79(11):1062-8. doi: 10.1016/j.lfs.2006.03.006. Epub 2006 Apr 21.; who are cite Aguirre-Crespo, F., Castillo-España, P., Villalobos-Molina, R., López-Guerrero, J.J., Estrada-Soto, S., 2005. Vasorelaxant effect of Mexican medicinal plants on isolated rat aorta. Pharmaceutical Biology 43 (6), 540–546; who are cite Villalobos-Molina R, Ibarra M (1996) a-Adrenoreceptorsmediating contraction in arteries of normotensive andspontaneously hypertensive rats are of the a1D or a1Asubtypes. Eur J Pharmacol 298: 257–263; who cite themselves and their foundations https://doi.org/10.1016/0014-2999(95)00074-U; https://doi.org/10.1016/0014-2999(93)90195-N; DOI: 10.1016/0014-2999(93)90195-n; 10.1016/s0922-4106(05)80019-7; ... and so on.
Somewhere in the quotation, there is a mention of preload 2 and 3g. However, if we focus on the animal type, we go from rabbit aorta to rat aorta. The first difference. The second difference is that I went through other publications and then used the citations of the author SRA. I found a line where he refers to previous work and at once I get to the 1g bias, as it should be.
Therefore, the question arises. Not on the basis of which publication, but on the basis of which scientific outputs was the bias set to 3g? I still think it's a lot, compared to 1-1.5g preload.
That is why I propose a consensus. Add either in the title or in the methods that it is an increased vascular pretension, but it would be appropriate to cite a publication that directly deals with tension in the vessel and differences in vasorelaxant and vasoconstrictor responses.
If this fact is pointed out, which you also emphasize in the limitation section of the manuscript, I am fully able to accept this fact.
RESPONSE: We agree with the Reviewer 1. Certainly, in the literature there is a large number of publications that mention that the optimal tension to observe the best response to the contraction or relaxation of the vascular tissue is between 1 and 3 g of initial tension, and that it remains constant until the blood vessel (in this case the thoracic aorta) is stabilized under ex vivo conditions. It is important to mention that the optimal tension is in that range, however, during the 20 years of experience that we have doing this experiment, we have found that for us, in our conditions and in our equipment, that the initial tension that allows a better stabilization and tissue stimulation is at 3 g, which gives us reliable results for the development of our research projects. Attached to this answer are 4 of our latest posts that support making this model.
- Ricardo Guzmán-Ávila, Samuel Estrada-Soto, Luis Arias-Durán, César Millán-Pacheco, Jaime Escalante-García, Maria Yolanda Rios, Virginia Flores-Morales, Rafael Villalobos-Molina, Gabriela Pérez-Barrón (2023). Vasorelaxant and antihypertensive effects of (3β)-ursen-12-en-3,28-diol by NO/cGMP system. Letters in drug design and discovery 20, 1959-1969.
- Samuel Estrada-Soto, Priscila Rendón-Vallejo, Rafael Villalobos-Molina, César Millán-Pacheco, Miguel A. Vázquez, Fernando Hernández-Borja, Emanuel Hernández-Núñez. (2022). 6-amino-3-methyl-4-(2-nitrophenyl)-1,4-dihydropyrano[2,3-c]pyrazole-5-carbonitrile shows antihypertensive and vasorelaxant action via calcium channel blockade. Drug Research 72, 53-60.
- Francisco Javier Aguirre Crespo, Elías Cerino Pérez, Janice D. G. Valdovinos Estrella, María G. Maldonado Velazquez, Benjamín O. Ortega Morales, Pedro Zamora Crecencio, Emanuel Hernández Núñez, Samuel E. Estrada Soto. (2021). Vasorelaxant and antioxidant activity of medicinal plants from Campeche, Mexico. Pharmacognosy Magazine 17 (73), 23-30.
- Luis Arias-Durán, Samuel Estrada-Soto, Monserrat Hernández-Morales, César Millán-Pacheco, Gabriel Navarrete-Vázquez, Rafael Villalobos-Molina, Maximiliano Ibarra-Barajas, Julio C. Almanza-Pérez (2021). Antihypertensive and vasorelaxant effect of leucodin and achillin isolated from Achillea millefolium through calcium channel blockade and NO production: functional ex vivo and in silico studies. Journal of Ethnopharmacology 273, 113948.
175-176 line: I am in favor of editing the text: we have control and add MESh. Relaxation looks to be 100% .... that is, MESH by way of ENDOTHELIUM/SMOOTH MUSCLE, the vascular segment relaxed.
By adding L-NAME or ODG, relaxation was present at 20%.
Response: The role of the vascular endothelium is affected by the presence of membrane-bound receptors like muscarinic, mainly M3 receptors, which when activated by an agonist induces nitric oxide production and release [28]. The MESh curve in the presence of atropine (an M3 antagonist) was not significantly modified. Thus, behavior discarded M3 receptor activation as the first trigger of the extract vasorelaxant effect. The major endothelium-derived releasing factor (EDRF) is nitric oxide (NO), which diffuses to smooth muscle cells where the guanylyl cyclase enzyme is activated, thus catalyzing GMP to cGMP conversion. To identify the participation of nitric oxide pathway activation, endothelium-intact aortic rings were pre-incubated with L-NAME (an NO synthetase inhibitor) or ODQ (a guanylyl cyclase inhibitor), respectively, and after that the MESh relaxant curve was obtained. Both curves were lowered in the presence of these inhibitors, thus suggesting that the NO-cGMP pathway could be involved in MESh-induced relaxation [29]. On the other hand, it is well known that PGI2 endothelium releases can evoke a vasorelaxant effect [30]. Thus, with the purpose to identified prostaglandins-participation in MESh effect, aortic rings were pre-incubated with indomethacin (an unspecific prostaglandins inhibitor); however, the MESh efficacy and potency were not modified, thus discarding PGI2-participation in MESh vasorelaxant effect [30].
Thus, the words "modification" and "may" are incorrect, but "sensitivity to the cumulative addition of MESh, after inhibition of the NA and sGC pathway, was reduced" (F2). At the same time, you observed that, unlike HESh and DESh, with MESh relaxation takes priority (as evidenced by a de-endothelialized vascular segment and a reduced ability to relax by 80%).
Line 190: How is curve 3a shifted to the right? Those are the tangents in the E50 that I mentioned last time
Response: As described EDRF can increase the second's messengers as cGMP or cAMP and thus cause PKG or PKA activation, which participates in the opening of KATP, potassium channel calcium-activation (KCa), and repolarization, in consequence the relaxation process is evoked [31]. In this context, the relaxant effect of MESh, in the presence of tetraethylammonium (KCa channels blocker), glibenclamide (an ATP-sensitive potassium channel blocker), and or 2-Aminopyridine (an inhibitor of KV), was significantly decreased by a half percent (Figure 3a), and the curve was shifted to the right respect to control (indicating a loss in potency), suggesting potassium channels opening because of NO-production MESh induced. The opening of potassium channels will cause cell repolarization, and consequently, calcium channel blocking occurs [32].
Line 234-235: You write about the use of the substances n-hexane, dichloromethane, and methanol, but no control of the pure substance, without active leached substances, is available in the publication or in the supplement. Either I didn't understand it, but you let the prepared substances evaporate. However, when there was, for example, methanol, was there no concentration of methanol in the resulting substance MESh? It would be good to emphasize and justify this, because, for example, methanol can react with the vascular segment. And you used 100% solutions according to the materials.
Response: All extracts obtained were subjected to evaporation (in a rotavapor) until totally dry. For pharmacological evaluation, the dry extracts were reconstituted in the appropriate vehicle to be added directly to the tissue.
line 268: camera or voltage sensor?
Response: Camera
